# Genotypes and phenotypes of G6PD deficiency among Indonesian females across diagnostic thresholds of G6PD activity guiding safe primaquine therapy of latent malaria

Ari Winasti Satyagraha[1]*, Arkasha Sadhewa[1], Lydia Visita Panggalo[1], Decy Subekti[1,2], Iqbal Elyazar[1,2], Saraswati Soebianto[1,2], Nunung Mahpud[1,2], Alida Rosita Harahap[1], J. Kevin Baird[1,2,3]

1 Eijkman Institute for Molecular Biology, Jakarta, Indonesia, 2 Eijkman-Oxford Clinical Research Unit, Jakarta, Indonesia, 3 Centre for Tropical Medicine & Global Health, Nuffield Department of Medicine, University of Oxford, Oxford, United Kingdom

* ari@eijkman.go.id

## Abstract

### Background

*Plasmodium vivax* occurs as a latent infection of liver and a patent infection of red blood cells. Radical cure requires both blood schizontocidal and hypnozoitocidal chemotherapies. The hypnozoitocidal therapies available are primaquine and tafenoquine, 8-aminoquinoline drugs that can provoke threatening acute hemolytic anemia in patients having an X-linked G6PD-deficiency. Heterozygous females may screen as G6PD-normal prior to radical cure and go on to experience hemolytic crisis.

### Methods & findings

This study examined G6PD phenotypes in 1928 female subjects living in malarious Sumba Island in eastern Indonesia to ascertain the prevalence of females vulnerable to diagnostic misclassification as G6PD-normal. All 367 (19%) females having <80% G6PD normal activity were genotyped. Among those, 103 (28%) were G6PD wild type, 251 (68·4%) were heterozygous, three (0·8%) were compound heterozygotes, and ten (2·7%) were homozygous deficient. The variants Vanua Lava, Viangchan, Coimbra, Chatham, and Kaiping occurred among them. Below the 70% of normal G6PD activity threshold, just 18 (8%) were G6PD-normal and 214 (92%) were G6PD-deficient. Among the 31 females with <30% G6PD normal activity were all ten homozygotes, all three compound heterozygotes, and just 18 were heterozygotes (7% of those).

### Conclusions

In this population, most G6PD heterozygosity in females occurred between 30% and 70% of normal (69·3%; 183/264). The prevalence of females at risk of G6PD misclassification as normal by qualitative screening was 9·5% (183/1928). Qualitative G6PD screening prior to

**Data Availability Statement:** All relevant data are within the manuscript and its supporting information files.

**Funding:** The study was funded by The Li Ka Shing Global Health Programme, grant number LG38 and JKB and AWS are the recipient of this grant. JKB, IE, DS, NM, and SS are supported by the Wellcome Trust Asia Africa Programme funding. AWS, AS, LVP, and ARH are supported by State Budget of Ministry of Research, Technology and Higher Education of the Republic of Indonesia. The funders had no role in study design, data collection and analysis, decision to publish, or preparation of the manuscript.

**Competing interests:** The authors have declared that no competing interests exist.

8-aminoquinoline therapies against *P. vivax* may leave one in ten females at risk of hemolytic crisis, which may be remedied by point-of-care quantitative tests.

## Author summary

*Plasmodium vivax* causes patent infection of red blood cells and latent infection of the liver. Radical cure for malaria effectively kills parasites in both blood and liver stages. Currently, radical cure for malaria involves either primaquine or tafenoquine, both of which cause acute hemolytic anemia in patients with an inherited defect in G6PD enzymatic activity. G6PD deficiency is an X-linked disorder and it is the most common enzyme deficiency in humans. Heterozygous females having one mutated and one normal gene may screen as G6PD normal in qualitative enzyme activity screening prior to primaquine therapy and be at risk of proceeding to hemolytic crisis. To date, there is no evidence-based G6PD activity cut-off value to distinguish those females who may not safely receive primaquine. This study aimed to inform this cut-off by a large survey of females by quantitative G6PD activity phenotyping along with genotyping of the G6PD gene. Two thousand females residing in a meso-endemic area in eastern Indonesia were screened for G6PD deficiency using qualitative and quantitative tests. Those with <80% G6PD activity of normal were genotyped. Among them, we found 0.3% were compound heterozygotes, 2.7% were homozygotes, 68.4% were heterozygotes for five variants of severe G6PD deficiency, and the rest (28.6%) were G6PD-normal. Applying a 70% cut-off excluded most of the G6PD-normals with relatively few G6PD-deficient females also being excluded. Our findings showed that 9.5% of the surveyed population would be at risk of misclassification as normal if using a qualitative test for G6PD deficiency. This study highlights the importance of quantitative G6PD screening of females living in a rural malarious area of Indonesia where G6PD prevalence is high and the variants are severe. Our evidence indicates a cut-off value of 70% of normal may be optimal for safe delivery of primaquine or tafenoquine therapies with minimal exclusion of those who may safely receive it.

## Introduction

Acute malaria caused by *Plasmodium vivax* asexual blood stage parasites provokes a debilitating febrile illness that may progress to severe and life-threatening disease syndromes associated with death [1,2]. More than 10 million of these attacks occur annually among the 2.8 billion people living at risk [3]. Each infection requires effective blood schizontocidal therapy to arrest progression of disease [4]. Unlike *Plasmodium falciparum*, the other major cause of human malaria, infection by *P. vivax* includes latency where dormant hepatic hypnozoites later awaken and provoke renewed attacks (relapses), each one threatening progression to severe disease and onward transmission. In cohorts living in endemic areas of Thailand and Papua New Guinea, hypnozoite-borne infections of blood accounted for >80% of incident *P. vivax* parasitemias [5,6]. Although the risk and timing of relapse varies across endemic zones [7], at least three and often five or more relapses per infectious event may be the rule [8]. In a cohort of 2,495 American soldiers repatriated from the Pacific theater of World War II, the median number of relapses over two years was 10–14 [9]. A study in eastern Indonesia followed tens of thousands of patients diagnosed with *P. falciparum* or *P. vivax* over a decade; while risk of death within 14 days was higher for *P. falciparum*, risk of multiple attacks, hospitalization, and premature death among *P. vivax* patients was two-fold higher [10]. Effective treatment of *P.*

*vivax* malaria that includes therapy against latency would thus provide very substantial clinical and public health benefits [11].

In contrast to the many therapeutic options for arresting the acute attack, the 8-aminoquinoline drugs primaquine or tafenoquine remain the only hypnozoitocidal options for terminating latency of *P. vivax* malaria. At therapeutic hypnozoitocidal doses, primaquine invariably provokes a potentially life-threatening acute hemolytic anemia in patients having glucose-6-phosphate dehydrogenase (G6PD) deficiency, an X-linked trait affecting over 400 million people at a rate of about 8% where endemic malaria occurs [12,13]. Many dozens of specific single nucleotide polymorphisms occur all across the 13 exons of the G6PD gene [14]. Among male hemizygotes and female homozygotes, G6PD impairment almost invariably occurs at <30% of normal enzyme activity and they are considered the most vulnerable to hemolysis caused by primaquine anti-relapse therapy [15]. The World Health Organization (WHO) recognized four classes of G6PD enzyme based on its activities: Class I as the most severe, Class II as severe, Class III as intermediate and Class IV as normal. Most variants are either class II or III.

Qualitative laboratory and point-of-care G6PD screening tests detect hemizygotes and homozygotes with nearly 100% sensitivity and specificity [16–19]. Female heterozygotes, however, may express G6PD activity phenotypes ranging from fully deficient to completely normal as a consequence of mosaicism of their red blood cell populations [20]. Apparently random inactivation of one X-chromosome or the other yields that heterogeneity of phenotype (a process called Lyonization that occurs during early embryonic development). Qualitative screening cannot reliably differentiate G6PD normal patients from those having >30% to <80% of normal enzyme activity [21,22]. This problem imposes uncertainty regarding a diagnostic reading of "normal" (and eligible for primaquine therapy) by those tests among females. Patients having G6PD activities between 30–80% are under-studied, but there have been reports of a drop in hematocrit up to 30% requiring transfusion therapy following high dose of primaquine, some of those experiencing hemolytic crises as severe as homozygous females [23,24].

In the current study we aimed to evaluate prevalent G6PD genotypes in relation to G6PD activity phenotype in a large sample of females living in a malaria-endemic area of Sumba Island in eastern Indonesia where we have conducted population surveys of G6PD heterogeneity [25,26]. This evidence directly informs the distribution of risk with G6PD screening in the context of primaquine therapy against latent malaria, and may serve to mitigate risk to G6PD heterozygotes. Reversing the historic neglect of this serious clinical and public health problem of poor access to hypnozoitocidal therapy should benefit males and females alike [27], and achieving that requires specific attention to the complexity of the heterozygous state of G6PD deficiency.

## Methods

### Ethics statement

A protocol detailing this survey was reviewed and approved by Eijkman Institute Research Ethics Committee (17th March 2015, EIREC Approval No. 81). Written informed consent was obtained from each participant and written informed consent was signed by guardian or parent of child participants.

### Study population

A cross-sectional prevalence survey of G6PD deficiency, anemia, and malaria status was undertaken in April 2015 at two sub-districts of Southwest Sumba: villages of Wainyapu and Waiha of Kodi Balaghar sub-district and Umbu Ngedo of Kodi Bangedo sub-district. In that

year, these locales had respective annual parasite incidences of 58·16, 85·24 and 26·85/cases per 1000 residents respectively according to local health district records (*P.falciparum* and *P. vivax*). Only healthy (without fever > 37.5˚C or history of fever within 48 hours) females older or equal to 6 years old who provided informed individual or parental/guardian consents were included in the survey. Based on our previous studies in Sumba Island, the prevalence of homozygous and heterozygous G6PD deficient females was 0.19 and thus to get the sample size we applied this equation:

$$n = \frac{Z^2 \times P \times (1 - P)}{d^2}$$

Z is the statistic corresponding to the level of confidence of 1.96 (95% confidence level) and d for precision (2%), our sample size was thus calculated:

$$n = \frac{1.96^2 \times 0.19 \times (1-0.19)}{0.02^2} = 1478$$

This value was 1345, and we increased that by 30% as a means of ensuring and adequate number heterozygous females for our research purposes, yielding a target sample size of 1921.

## Venipuncture and malaria screening

Willing subjects who signed written informed consent were asked to allow taking 3 ml of venous blood collected in EDTA tubes. Residual blood in the syringe was used for microscopic examinations for malaria parasites (thick and thin blood smears) in the field laboratory according to standard protocol by technicians certified as competent according to WHO specifications in doing so. Participants found to be positive for malaria infection were offered treatment with therapies as stipulated by the Indonesian Ministry of Health guidance the next day. Blood in sealed EDTA tubes was put on ice immediately after venipuncture and stored at 4˚C within a few hours. Within 3 days these samples were transported on cold packs by air to Jakarta for the laboratory analyses detailed here.

## G6PD qualitative and quantitative tests

G6PD qualitative testing was done at point-of-care using the Carestart G6PD rapid diagnostic test (CSG, Accessbio, USA) according to the manufacturer's protocol. In brief, blood placed into test cassette with liquid reagents migrates across a white cellulose wick within a maximum of minutes: G6PD-normal blood causes a distinct purple color to develop, whereas with G6PD-deficient blood either remains white or develops only a very slight purple hue. Technicians performing the test were instructed and trained to classify no color or only lightly colored tests as G6PD-deficient. Refresher training for this took place for 2 days prior to going to the field. Most CSG were read by single technician. A second technician was conferred when the result was found difficult to interpret.

Blood samples in Jakarta were held at 4˚C and analyzed less than 24hr arrival from Sumba. These were first examined for hemoglobin (HemoCue Hb 201, Sweden) according to the manufacturer's instruction. Samples having hemoglobin measurements of less than 10 g/dL were excluded from further analyses in order to avoid the impact of anemia causing falsely elevated G6PD readings [22].

G6PD quantitative test from Trinity Biotech (Cat # 345-B) was used as the reference test for G6PD activity screening. The test relies on the principle of NADPH formation that is proportional to the G6PD activity directly measured using a spectrophotometer (Shimadzu UV-1800 series) at 340 nm absorbance, precisely as prescribed by the test manufacturer. Briefly, 10 μl of

blood was added to a snap cap tube containing 1.0 ml of G-6-PDH Assay Solution, mixed thoroughly and incubated at 30˚C for 5–10 min before adding 2.0 ml of G-6-PDH Substrate Solution. The tube and was mixed gently by several inversions. The contents were transferred to a cuvette and incubated in the temperature-controlled spectrophotometer at 30˚C for five min. The cuvette was then read at 340 nm using water as blank and marked as INITIAL A. The reaction was incubated for another five min at 30˚C where the contents were read again and marked as FINAL A. G6PD activity was then calculated as U/g Hb by subtracting FINAL A with INITIAL A and divided by five as the manufacturer's manual instructs. The reading was done once per sample. The absolute value for normal G6PD activity (as 100%) was calculated as the mean of G6PD activity among subjects having ≥5U/g Hb.

## G6PD genotyping

G6PD genotyping was restricted to samples showing <80% of normal G6PD activity. DNA was extracted from EDTA blood samples using a modified salting out method as previously described [28]. Extracted DNA was amplified in PCR and digested with specific restriction enzymes for common variants (Vanua Lava, Viangchan, Chatham, Coimbra and Kaiping) in these regions [25,26]. Samples showing no digestion by those enzymes were whole-gene or next-generation sequenced using primers described by Saunders et al [29].

## Statistical analysis

We described the female sample characteristics including age median and range, hemoglobin measurements, G6PD point-of-care screening results (G6PD deficient, G6PD normal), and malaria microscopic findings (negative/positive for malaria, *Plasmodium* species). The primary outcome was the proportion of G6PD deficient females in the female sample population. We analyzed G6PD enzyme activities using quantitative G6PD Trinity Biotech for the samples with hemoglobin values > 10 g/dl. Mean, median, standard deviations and range of G6PD enzymatic activities were calculated to determine the reference values in normal and deficient subjects. We assessed 100% G6PD activity as the mean of the G6PD activity ≥5U/g Hb as our cut off, and set our 30%, 70% and 80% diagnostic thresholds based on this value. The 5 U/g Hb cut-off was considered the lowest limit of truly normal G6PD activity and protected the estimate of 100% enzyme activity from the diminishing bias of G6PD-deficiency. The diagnostic thresholds reflected those considered effective for CSG (<30%) or safe for administration of 8-aminoquinolines (>70% or >80%, depending on authoritative recommendations) [25,26] We classified genotype characteristics (wild type G6PD genotype, heterozygotes, compound heterozygotes, and homozygotes deficient) for those females with G6PD deficiency. We measured the performance of CareStart G6PD rapid test specific to genotypes and G6PD activity ranges among female subjects below 80% of normal. We also measured sensitivity, specificity, negative and positive predicted value of CSG against spectrophotometric reference test at 10%, 30%, 70% and 80%. A 95% confidence of each indicator was measured. Statistical significance of G6PD prevalence was evaluated by Fisher's exact test. Scattered boxplot was used to show how the G6PD activities are within G6PD variants identified in the study. Data were analyzed using R software.

## Results

### Field survey

A total of 2056 females were screened. Table 1 lists their demographic characteristics along with G6PD point-of-care screening, hemoglobin measurements, and malaria blood film examination findings. The prevalence of microscopy positivity for malaria was 6·7% (129/1928),

**Table 1. Survey sample demographics and screening findings.**

| | |
|---|---|
| Sample size | 2056 |
| Age in years (median) * | 22·5 (95%CI, IQR 24) |
| Age range (years)* | 6–90 |
| Subjects having Hemoglobin ≥10 g/dL | 1928 |
| Subjects having Hemoglobin<10 g/dL | 128 |
| Subjects having G6PD qualitative test NORMAL | 1814 |
| Subjects having G6PD qualitative test DEFICIENT | 113** |
| Subjects Negative for malaria *# | 1799 |
| Subjects Positive for malaria*# | 129 |
| Numbers of cases of microscopically patent infection by *P. falciparum/P. vivax/P. malariae/* Mixed species* | 78/38/1/12 |

*these are calculated from samples Hb>10 g/dL. All other entries with exception of Age are in actual number of persons.

**one sample did not have qualitative test result.

# malaria diagnostic was from microscopy only.

with *P. falciparum* being the dominant species by a margin of 2:1. *P. vivax* was found in 53 females, including 11 of 12 mixed species infections (prevalence of 2·6%). A total of 1928 females (93·8%) had hemoglobin levels ≥10g/dL and were included in the study (Fig 1). The point-of-care G6PD rapid test identified 1814 women as normal and 113 women as deficient (1 sample did not have CSG data), indicating a prevalence of 6·2% for G6PD deficiency using that device.

## Laboratory findings

Quantitative G6PD testing was accomplished for 1928 samples, with all 128 exclusions being due to hemoglobin values below 10g/dl (Fig 1). Fig 2 illustrates the frequency distribution of subjects across the range of observed G6PD activity values. The mean value among those ≥5 U/g Hb was 11·04 U/g Hb and set as 100% of normal enzyme activity. The thresholds of 10%, 30%, 70%, and 80% are illustrated in Fig 2. The distribution of malaria-positive subjects by species of diagnosis, also in Fig 2, appeared independent of G6PD activity level.

Fig 3 illustrates G6PD measurements among the 367 subjects having <80% of G6PD normal and according to genotype. Vanua Lava and Viangchan variants co-dominated, with Chatham, Coimbra, and Kaiping variants in minority representation. The very few homozygotes (10) were represented by Vanua Lava, Viangchan, and Coimbra, and all occurred at the lowest spectrum of G6PD activity. The 3 compound heterozygotes represented by Viangchan/Vanua Lava and Chatham/Vanua Lava also occurred at the lower end of G6PD activity (<2 U/g Hb). The majority of G6PD deficiency among females occurred as heterozygotes having between 30% and 70% of normal activity, with few exceptions below the lower threshold and relatively more above the higher threshold; whereas 103 G6PD-normal subjects (Normal and ND) occurred below the 80% threshold, just 18 did so below the 70% threshold. The ND (not determined) lane of Fig 3 represents subjects with <80% of normal G6PD activity negative for RFLP analysis for the SNPs analyzed but who were not whole gene sequenced. Excepting the two subjects below 5 U/g Hb, ND subjects were presumed G6PD-normal.

The activity values among heterozygotes appeared normally distributed. Among the 251 heterozygotes detected in this survey, 73% (183) occurred between 30% and 70% of G6PD normal activity, within ±1 standard deviation of the 50% mean predicted for a random

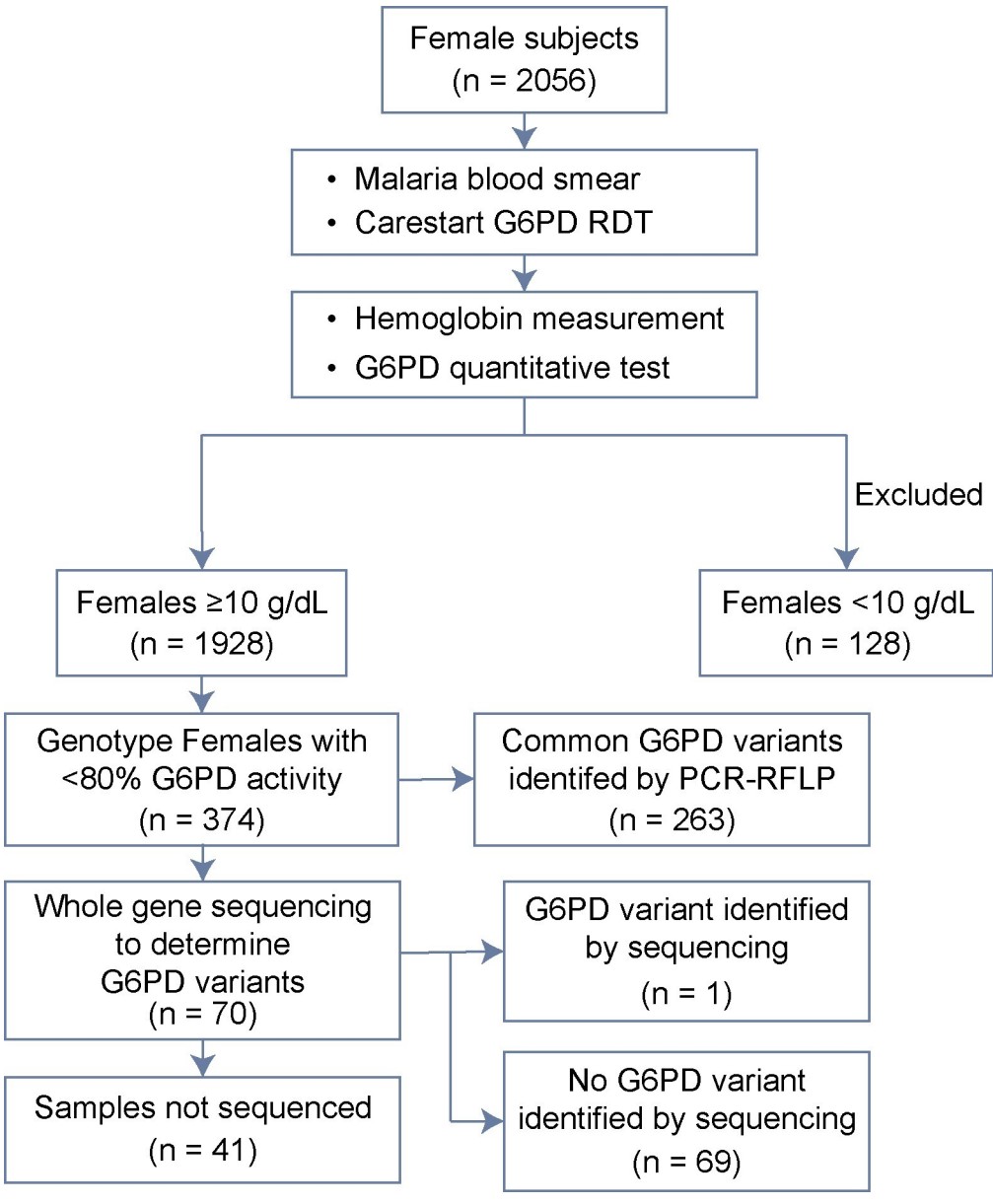

**Fig 1. Survey process and sample analysis.**

distribution between 0 and 100%. Fig 4 illustrates the distribution of G6PD genotypes between 0 and 70% of normal G6PD activity. The relative predominance of G6PD-normal subjects (Normal and ND) above the 70% threshold and paucity of the same below that threshold may be seen. Heterozygotes overwhelmingly dominated the 30% to 70% range. Likewise, below the 30% threshold, homozygotes and compound heterozygotes dominated the lowest values therein, with most of the heterozygotes just below 30%.

### Diagnostic result of CareStart G6PD rapid test

Table 2 summarizes the essential features of diagnostic result of the CareStart G6PD rapid test (CSG) relative to the defining Trinity quantitative spectrophotometric assay and genotype

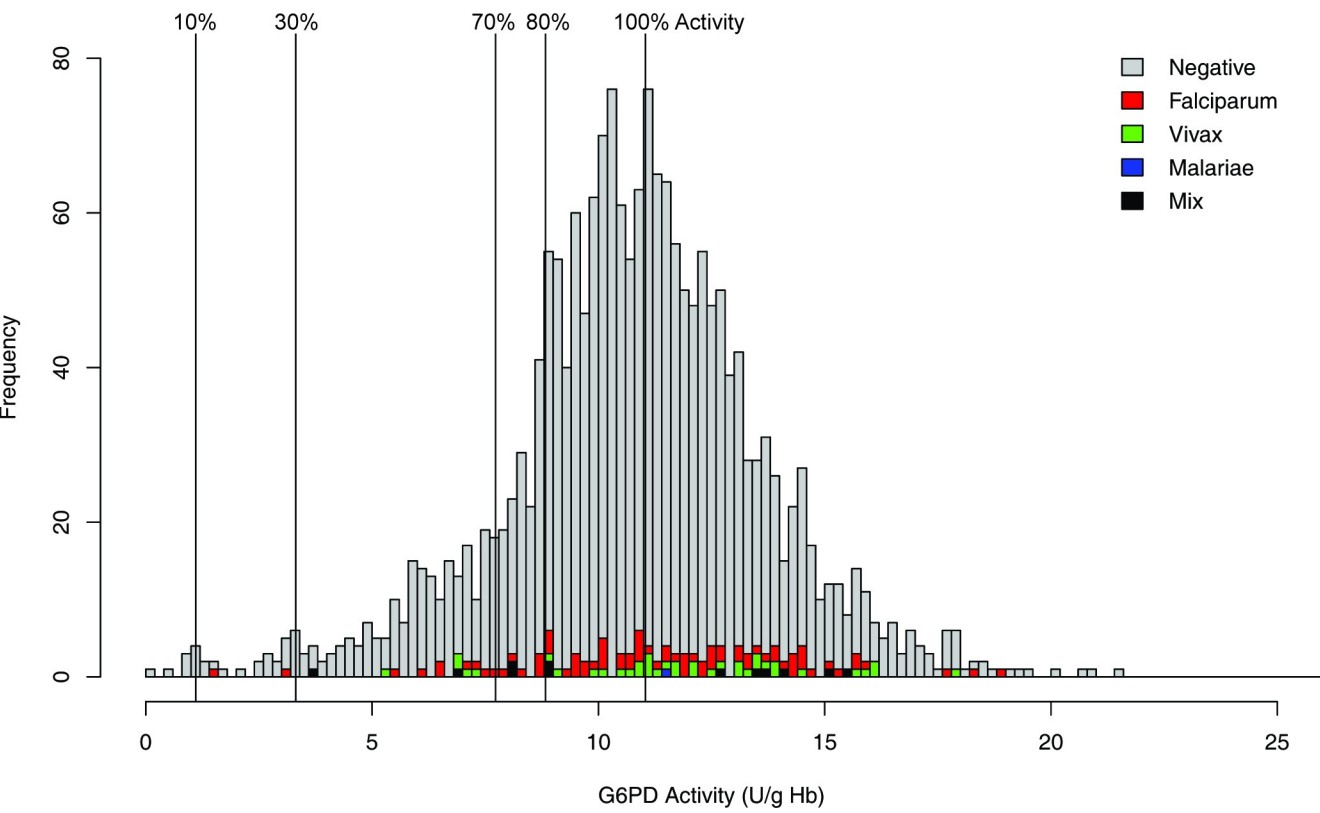

**Fig 2. G6PD activity distribution and malaria microscopy positivity in the survey.**

result among the 1928 non-anemic subjects (>10 gHb/dL) at varied thresholds of G6PD activity (% of normal). The CSG detected only 33 of 183 (18·0%) heterozygotes between 30% and 70% of normal G6PD activity as G6PD-deficient. The overall sensitivity, specificity, positive predictive value, and negative predictive value of the CSG relative to quantitative testing at the 70% of normal activity threshold was 25.1% (95%CI, 19.7–31.2), 96.8% (95%CI, 95.8–97.6), 21.4% (95%CI, 16.2–27.7), and 97.4% (95%CI, 97.2–97.5), respectively.

## Discussion

The findings of this survey in malarious Sumba Island in eastern Indonesia affirms a G6PD activity threshold of 70% of normal in identifying most G6PD-deficient females for exclusion from 8-aminoquinoline therapies with minimal exclusion of G6PD-normal females. Among the 232 subjects below that threshold, 214 (92%) had genotype-confirmed G6PD deficiency. At a threshold of 80% of normal activity 264 out of 367 (72%) subjects were G6PD-deficient. The 50 G6PD-deficient subjects having <80% but >70% represented 19% of G6PD-deficient females present in the sample. Given the relatively high degree of enzyme activity among those females, which reflect a high proportion of G6PD-normal red blood cells, they presumably represented the least vulnerable to 8-aminoquinoline hemolytic crisis among G6PD-deficient females. While an 80% threshold would exclude those 50 potentially susceptible females from therapy, it would also have excluded 103 G6PD-normal women representing 5% of the population as a whole. The unnecessary exclusion of G6PD-normal women from 8-aminoquinoline therapy is reduced to 0.9% of the population at a threshold of 70%. The actual safety threshold of G6PD activity for anti-relapse therapy with primaquine or tafenoquine has not been

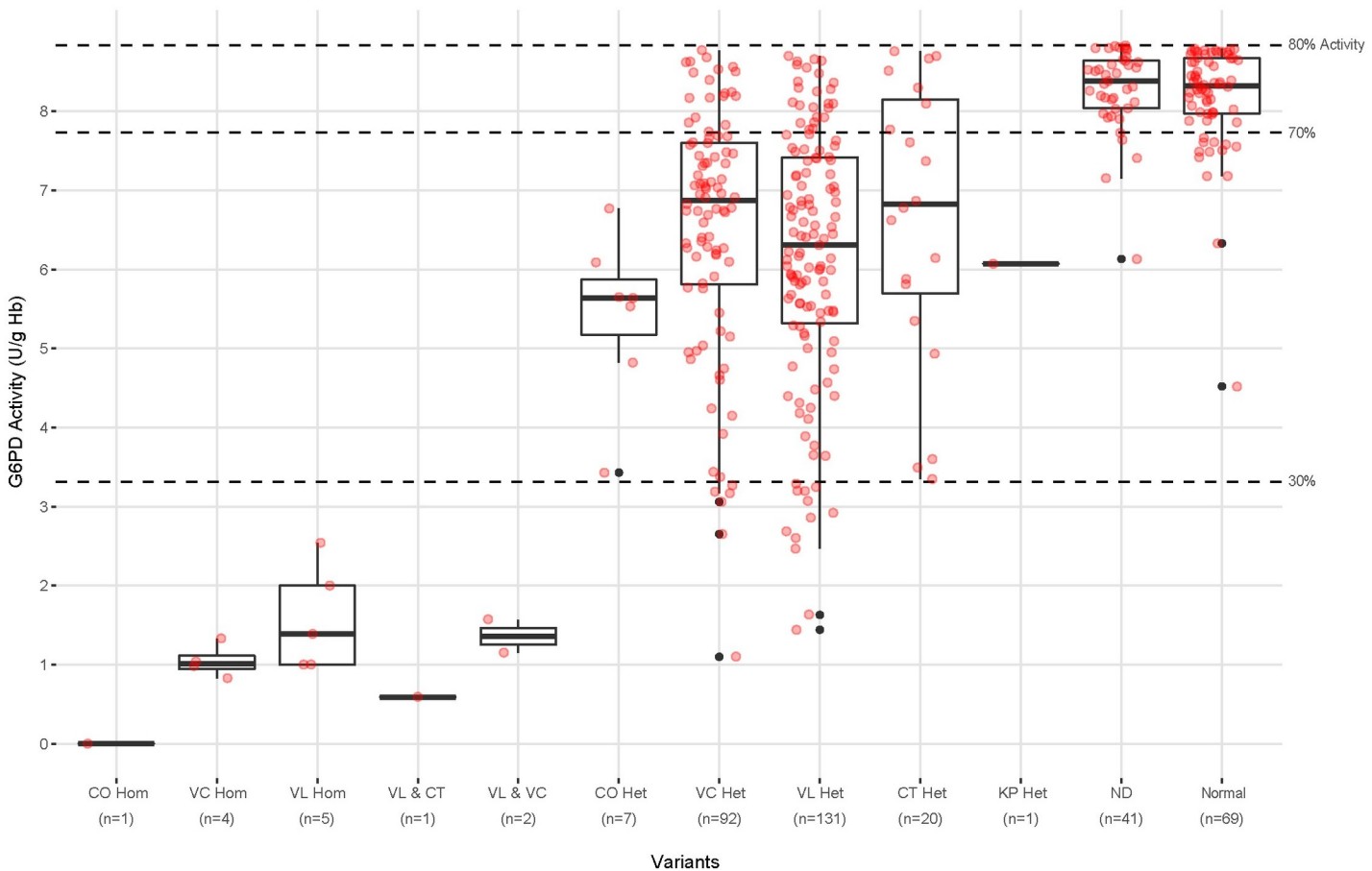

**Fig 3. Genotypes identified among the subjects having <80% of normal G6PD activity and that phenotype illustrated in relation to 30% and 70% of normal activity thresholds.** CO, VC, and VL Hom are Coimbra, Viangchan, and Vanua Lava homozygous respectively. CO, VC, CT and VL Het are Coimbra, Viangchan, Chatham and Vanua Lava heterozygous respectively. VL & CT and VL & VC represent the 3 compound heterozygotes of those variants. The ND lane represent subjects negative for the G6PD variant SNPs analyzed, but not whole gene sequenced, whereas the Normal lane represents those whole gene sequenced as wild type G6PD. The black dots represent outliers calculated from the boxplot whereas the pink outliers are calculated from the scatterplot of same samples.

determined in practice or clinical research on heterozygotes. Any threshold given remains a supposition and our findings do not offer the assurance of safety at a 70% activity threshold. Instead, our findings only inform the distribution of G6PD heterozygotes at these thresholds of diagnostic performance (<30%) or those of safety (70% or 80%).

As elsewhere in much of the malaria endemic world, most acute malaria in eastern Indonesia occurs at the under-resourced periphery of healthcare delivery. That setting may not accommodate sustainable quantitative G6PD assessments allowing the application of a 70% threshold defining 8-aminoquinoline therapy *proceed* vs. *do not proceed* rules. We assessed the likely impacts of qualitative vs. quantitative G6PD assessments in the Sumba population of females surveyed. As reported by others and described from our laboratory [30,31], the CSG qualitative test becomes increasingly insensitive to deficiency at G6PD activities above 30% as it is not designed to distinguish 30–80% from >80% individuals. The sensitivity, specificity, negative and positive predicted values at cut off 70% were 25.1%, 96.8%, 21.4% and 97.4% respectively. The sensitivity and positive predicted values further dropped at 80% cut off (S1 Table and S1 Fig). The current survey showed that 82% of the 183 G6PD-deficient females having G6PD activities between 30% and 70% screened as G6PD-normal by the CSG. In other

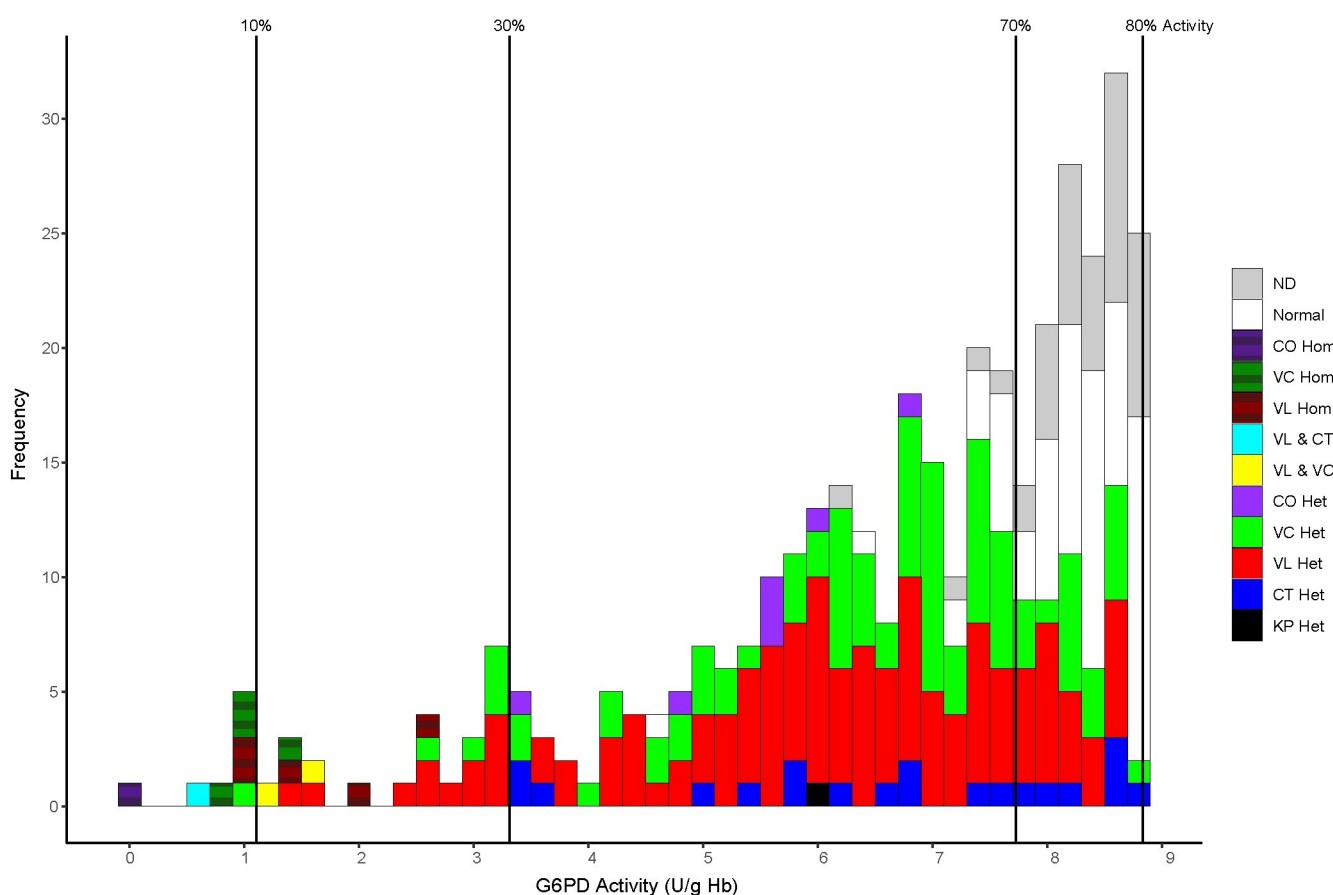

**Fig 4. Distribution of G6PD genotypes among phenotypic activity thresholds.** ND-not determined, CO–Coimbra, VC–Viangchan, VL–Vanua Lava, CT–Chatham, and KP–Kaiping. Hom–homozygous mutant and Het–heterozygous. VL & CT, and VL & VC represent the compound heterozygotes.

words, 9·5% (183/1928) of all females at the study sites on Sumba would be at high risk of being cleared for potentially dangerous 8-aminoquinoline therapies with qualitative G6PD screening.

Varied national drug regulators have registered tafenoquine as a single-dose alternative to the standard 14 days daily dosing with primaquine for presumptive anti-relapse therapy (PART) following a diagnosis of *P. vivax* malaria [32,33]. The administration of tafenoquine is not recommended in patients having <70% of normal G6PD activity; thus, imposing quantitative testing as a necessity corroborated by our findings from females residing in Sumba.

**Table 2. Summary of diagnostic performance of the CSG specific to genotypes and G6PD activity ranges among individual subjects below 80% of normal.**

| Genotype | Percent of Normal G6PD Activity Threshold | | | | | |
|---|---|---|---|---|---|---|
| | 71–80% | | 30–70% | | <30% | |
| | # Genotype | # CSG Deficient | # Genotype | # CSG Deficient | # Genotype | # CSG Deficient |
| Homozygous | 0 | 0 | 0 | 0 | 10 | 8 (80%) |
| Compound Heterozygous | 0 | 0 | 0 | 0 | 3 | 3 (100%) |
| Heterozygous | 50 | 4 (8%) | 183 | 33 (18%) | 18 | 14 (77.8%) |
| Normal | 85 | 2 (2.4%) | 18 | 0 | 0 | 0 |
| TOTAL | 135 | 6 (4.4%) | 201 | 33 (16.4%) | 31 | 25 (80.6%) |

Indeed, a Kenyan female A- (Class III) heterozygote accidently dosed with 400mg tafenoquine required transfusion therapy [34]. Among healthy ethnic Karin women in Thailand heterozygous for the moderate Class III Mahidol variant of G6PD deficiency and having between 40% and 60% of normal G6PD activity, all three subjects receiving the 300mg single dose of tafenoquine hemolyzed approximately 8% of their red blood cells [35]. All of the five variants identified at Sumba (Vanua Lava, Viangchan, Chatham, Coimbra, and Kaiping) were Class II and perhaps more vulnerable to 8-aminoquinoline toxicity than Mahidol Class III variant.

Qualitative G6PD screening prior to PART applying low-dose primaquine (0·25mg/kg/d x 14d) may be viewed as adequately safe for females because the prolonged dosing allows mitigation of potential harm by cessation of therapy after the onset of symptoms of acute hemolytic anemia. However, that onset occurs late and abruptly; typically, a day after the 3rd or 4th dose with a sudden drop of hemoglobin accompanied by haemoglobinuria and jaundice, i.e., with hemolytic crisis in progress [36,37]. There is no validated means of monitoring for those events in averting serious harm, i.e., specification of any specific clinical or laboratory parameter at a designated time point during treatment reliably indicating cessation of treatment before serious harm is done. In eastern Indonesia severe Class II variants are both dominant among G6PD-deficient people and highly prevalent in the general population [38]. Dosing G6PD unknowns, especially when using the more efficacious high-dose regimens of primaquine (0·5mg/kg/d x 14d or 1·0mg/kg/d x 7d), cannot be recommended at sites like Sumba without quantitative test screening.

Primaquine efficacy is further complicated by both impacts of partner blood schizontocides and natural polymorphisms of cytochrome P450 2D6 impairing the necessary metabolic processing of primaquine [39–42]. Inadequately treated *P. vivax* latency in Sumba—where most people at risk of malaria live in impoverished rural settings with chronic malnourishment and co-endemic neglected tropical infections–likely exerts the same insidiously harmful effects observed elsewhere in eastern Indonesia [43]. Furthermore, studies [44,45] have shown the increased risk of anemia due to relapses in no or low dose (0·25 mg/kg) of primaquine. Access to efficacious primaquine therapy against latent malaria at sites like Sumba will require either clinically validated cease-dosing criteria or quantitative G6PD screening. Currently, there are several point-of-care quantitative G6PD screening tests available, but none has yet been validated as reliable and practical for lay users in village settings. In Indonesia during 2021, we found one such commercially available kit costs USD 824 for the instrument, USD12 per test strip and USD 104 for controls, with promising performance that may be used in these instances *in lieu* of reference test [46,47]. However, the high cost per strip and 1 year expiration date may hinder its wide use in Indonesia.

Primaquine has been in clinical use for nearly 70 years despite of its hemolytic toxicity in G6PD-deficient patients. Global health authorities and national malaria control programmes alike advise using primaquine to prevent relapse of latent malaria applying varying doses, dosing strategies, and G6PD precautions [48]. As the findings reported here illustrate, the therapeutic dilemma of 8-aminoquinoline anti-relapse therapy is complex and involves significant proportions of local populations being vulnerable to potential harm. Protecting the vulnerable minority by withholding therapy exposes all infected to the harm of repeated acute attacks. The health authorities in eastern Indonesia, as in many endemic areas, cannot confidently attack the latent reservoir of that harm because rational and practical guidance for safely and effectively doing so is lacking. The findings reported here aimed to aid in formulating that guidance with specific respect to G6PD screening parameters that help to mitigate risk of hemolysis by 8-aminoquinolines. Practical guidance for efficacious radical cure without quantitative G6PD testing may require validated parameters of clinical monitoring for acute hemolytic anemia.

## Study limitation

This study is limited in the biochemical aspect since it did not compare the Kcat of G6PD from genotypically G6PD normal and heterozygous females with G6PD activities below 80%. This knowledge on top of G6PD genotype and activity can help fine tune the hemolytic risk of G6PD heterozygous females in oxidative stress.

## Supporting information

**S1 Table. Performance of CSG G6PD Test against reference test Pointe Scientific quantitative assay.**
(DOCX)

**S1 Fig. CSG G6PD results relative to G6PD activities derived from the reference test. Red boxes are deficient results by CSG and blue boxes are normal results by CSG.**
(TIF)

## Acknowledgments

We thank the participating subjects at villages Wainyapu, Waiha and Umbu Ngedo. The authors thank Dedi Sudiana as field coordinator, Sunardi as field microscopist, Yustus Selan as the Southwest Sumba Regency Malaria Program Officer, Lisa Mone and Nathan as Malaria Program Officers at Panenggo Ede Health Center who helped during field sampling. The authors thank Rosalie Elvira, Jeny, Lia Waslia and Denny Feriandika for their technical expertise in the lab/field and Fitria Wulandari for her excellent administrative help. The authors also thank Dr. Bob Taylor and Prof. Lucio Luzzatto for their thorough critical review of this manuscript.

## Author Contributions

**Conceptualization:** Ari Winasti Satyagraha, J. Kevin Baird.

**Data curation:** Ari Winasti Satyagraha, Arkasha Sadhewa, Iqbal Elyazar.

**Formal analysis:** Ari Winasti Satyagraha, Arkasha Sadhewa, J. Kevin Baird.

**Funding acquisition:** Ari Winasti Satyagraha, J. Kevin Baird.

**Investigation:** Ari Winasti Satyagraha, Arkasha Sadhewa, Lydia Visita Panggalo, Decy Subekti, Saraswati Soebianto, Nunung Mahpud, Alida Rosita Harahap.

**Methodology:** Ari Winasti Satyagraha, Lydia Visita Panggalo, Decy Subekti, Saraswati Soebianto, Nunung Mahpud, Alida Rosita Harahap.

**Project administration:** Ari Winasti Satyagraha, Decy Subekti, Saraswati Soebianto, Nunung Mahpud.

**Resources:** Ari Winasti Satyagraha, Decy Subekti, Saraswati Soebianto, J. Kevin Baird.

**Software:** Arkasha Sadhewa, Iqbal Elyazar.

**Supervision:** Ari Winasti Satyagraha, Decy Subekti, Alida Rosita Harahap, J. Kevin Baird.

**Validation:** Ari Winasti Satyagraha, Lydia Visita Panggalo, Iqbal Elyazar, Alida Rosita Harahap, J. Kevin Baird.

**Visualization:** Ari Winasti Satyagraha, Arkasha Sadhewa, Lydia Visita Panggalo, Iqbal Elyazar, J. Kevin Baird.

**Writing – original draft:** Ari Winasti Satyagraha.

**Writing – review & editing:** Ari Winasti Satyagraha, Arkasha Sadhewa, Lydia Visita Panggalo, Decy Subekti, Iqbal Elyazar, Saraswati Soebianto, Nunung Mahpud, Alida Rosita Harahap, J. Kevin Baird.

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
