## [Decision Letter · Decision Letter 0]

4 May 2021

Dear Dr Satyagraha,

Thank you very much for submitting your manuscript "G6PD Deficiency Among Indonesian Females Across Diagnostic Thresholds for Safe Primaquine Therapy of Plasmodium vivax Malaria" for consideration at PLOS Neglected Tropical Diseases. As with all papers reviewed by the journal, your manuscript was reviewed by members of the editorial board and by several independent reviewers. In light of the reviews (below this email), we would like to invite the resubmission of a significantly-revised version that takes into account the reviewers' comments. 

We cannot make any decision about publication until we have seen the revised manuscript and your response to the reviewers' comments. Your revised manuscript is also likely to be sent to reviewers for further evaluation.

Sincerely,

Wuelton Marcelo Monteiro, Ph.D.

Associate Editor

Mary Lopez-Perez

Deputy Editor

Reviewer's Responses to Questions

**Key Review Criteria Required for Acceptance?**

**Methods**

-Are the objectives of the study clearly articulated with a clear testable hypothesis stated?

-Is the study design appropriate to address the stated objectives?

-Is the population clearly described and appropriate for the hypothesis being tested?

-Is the sample size sufficient to ensure adequate power to address the hypothesis being tested?

-Were correct statistical analysis used to support conclusions?

-Are there concerns about ethical or regulatory requirements being met?

Reviewer #1: The methods are appropriate and adequately described.

Reviewer #2: Are the objectives of the study clearly articulated with a clear testable hypothesis stated?

• While the objectives are clearly formulated, the article does not present a hypothesis (not needed due to observational character).

Is the study design appropriate to address the stated objectives?

• Yes

Is the population clearly described and appropriate for the hypothesis being tested?

• It is unclear how the study population was selected? Kindly add details on the cross-sectional survey and how eligible females were enrolled.

• The authors state that only “healthy” females were enrolled. How was this determined? Was body temperature and history of fever considered? 

Is the sample size sufficient to ensure adequate power to address the hypothesis being tested?

• The authors did not include a power calculation – kindly add.

Were correct statistical analysis used to support conclusions?

• The description of statistical methods is insufficient:

o The definition of 100% G6PD activity is not explained in the methods, but only in the results, kindly add.

o The authors have defined 100% activity as the mean activity of all females with activities above 5U/gHb – kindly justify this method. From figure 2 I would have thought 100% activity is a bit lower, accordingly 30%, 70% and 80% need to be lower as well, the definition of 100% activity has obvious consequences for the conclusion

o Rather than stating numbers of erroneously categorized individuals by Carestart RDT, kindly present sensitivity, specificity, NPV and PPV => discuss these findings in the discussion

o There was a mean / median / max. (?) delay between sample collection and spectrophotometry and this delay will have impacted measured G6PD activity. Can the authors analyse the impact the delay has had (regression analysis) and if needed correct their findings accordingly? This may alter the discussion and conclusion.

Are there concerns about ethical or regulatory requirements being met?

• The authors state that “willing” females were enrolled. Can you kindly clarify how willingness to participate was documented?

Reviewer #3: (No Response)

**Results**

-Does the analysis presented match the analysis plan?

-Are the results clearly and completely presented?

-Are the figures (Tables, Images) of sufficient quality for clarity?

Reviewer #1: The results are clearly and completely presented.

Reviewer #2: Does the analysis presented match the analysis plan?

• Yes

Are the results clearly and completely presented?

• No

o The authors state a malaria prevalence of 7%, kindly add absolute numbers, overall as well as for each species. Since participant identification is not explained it is unclear how reflective this prevalence is of the overall population.

o The sentence in line 194 – 195 “A total of 1928 females…” needs to go before the sentence “The point of care G6PD rapid…” (line 192 – 193). Otherwise the reader is surprised that not everybody enrolled was tested

o The authors state that 1814 females were G6PD normal and 113 females deficient (total = 1927), however also state that 1928 females were tested, there seems to be one result missing? 

o If possible, provide a table with some baseline data on the study population

o Table 1 contains the row: “Age (median)” and next to it the result: “22.5 (95%CI)”. Please state median and interquartile range (IQR)

o Table 1: Please state that all other entries refer to total number of persons

o The authors state that 103 females (approx. 5% of the study population) were G6PD normal by genotype but had activities of less than 80% - this could suggest that the definition of 100% is not correct (too high)? Can the authors discuss this?

Are the figures (Tables, Images) of sufficient quality for clarity?

• No, see comments on table 1 above, replace contents of table 2 with standard performance indicators (and 95%CI) and prevalence at the respective cut offs.

Reviewer #3: (No Response)

**Conclusions**

-Are the conclusions supported by the data presented?

-Are the limitations of analysis clearly described?

-Do the authors discuss how these data can be helpful to advance our understanding of the topic under study?

-Is public health relevance addressed?

Reviewer #1: The conclusions are appropriate.

Reviewer #2: Are the conclusions supported by the data presented?

• No, the authors base their conclusions on a questionable definition of 100% G6PD activity:

o There was a delay of up to 3 hours before samples were introduced to a 4oC environment, G6PD activity deteriorates at room temperature (specifically in the tropics) rapidly and this delay will have introduced a significant bias, specifically if the delay varied

o Delay to testing was up to 13 days and this will have further affected measuredG6PD activity

o The definition of 100% activity is not justified 

… in consequence and in the absence of any clinical data it does not seem justified to infer these findings to the impact a cut off of 70% or 80% activity have on patient safety when treated with radical cure in this setting. 

• The authors conclude that point of care testing for heterozygous females within the setting is challenging if not impossible, but do not mention novel G6PD biosensors that could address this diagnostic gap. Can you kindly reflect on this?

• The authors also conclude that a qualitative diagnostic is not suitable to guide radical cure in heterozygous females, and this is correct, however has been described earlier.

Are the limitations of analysis clearly described?

• A limitations section is missing

Do the authors discuss how these data can be helpful to advance our understanding of the topic under study?

• yes

Is public health relevance addressed?

• Yes

Reviewer #3: (No Response)

**Editorial and Data Presentation Modifications?**

Reviewer #1: Title

It is difficult to understand from the title what the paper is about. Perhaps the title could be revised so as to be more informative?

Abstract: The conclusions point to the deficits of qualitative tests. Perhaps the abstract could end of a more positive note highlighting the ongoing roll-out of quantitative tests?

Intro: Please consider rephrasing the following sentence? “This

113 evidence directly informs the distribution of risk with G6PD screening in the context of primaquine

114 therapy against latent vivax malaria, knowledge enabling rational strategies for maximizing essential

115 safe access to treatment against a highly prevalent and dangerous infection.”

Table 2: Please explain what you mean by double heterozygous? Please explain what is CSG?

Discusssion: 

“The administration of tafenoquine is prohibited in

289 patients having <70% of normal G6PD activity;”

“prohibited” may not be the most appropriate term? Better say “not recommended”.

Reviewer #2: • Parts of the manuscript need major revision for spelling, grammar, clarity and consistency. 

• Introduction and discussion are quite long

Reviewer #3: Line 57: replace “chromosome” by “gene”

Line 79: replace “case” by “species” or “cause”

Line 96: “about 8%”: this is a global average, but very variable between populations

Line 102: could also cite https://journals.plos.org/plosmedicine/article?id=10.1371/journal.pmed.1002992

Line 115: “highly prevalent”: citation needed

Line 124: “annual parasite incidence..” specify whether all species

Line 125: suggest “at least 6 yrs” to avoid confusion with Table 1.

Line 133: if possible, state by which authority/system microscopists were certified?

Line 136: “within a few hours”: please be more specific

Line 137: “within 3 days”: specify that this is the transport (not the analyses) eg.. “then cold-transported to Jakarta within 3 days for …”

Line 139: any details available on test version number? Product ID?

Line 141: how many minutes?

Line 143: details on training, previous experience? Were RDTs read by single technician, or double read?

Line 145: provide citation/justification of sample integrity after 10 days for conducting enzyme activity studies

Line 155: was spectrophotometry performed in duplicate/replicate?

Line 180: suggest double or compound heterozygotes: suggest keeping consistent

Line 193: specify G6PD rapid test (not previously referred to RDT)

Table 1: include units; specify that pos/neg for malaria is by microscopy

Line 204: ≥5 U/g Hb stated in the methods (line 161)

Line 207: did authors test this? Visual assessment suggests may have been more Pv among those with >100% activity

Line 209: confirm figures, Fig 1 suggests n=374?

Line 210: “vanua lava and viangchan variants co-dominated”: possible to quantify?

Line 214: “lower end of G6PD activity”: perhaps could say <3 Ug/Hb?

Line 229: could provide more detail on the outliers in the Methods

Line 235: Fig 4 shows up to 80% (not 70%)

Line 233: could specify “among the 251 heterozygotes” with <80% normal activity.

Line 250: interesting result that could be mentioned is the cases with <30% activity not identified by the CSG. Could consider including another histogram like fig 4 to plot out the CSG results against activity levels. 

Table 2: suggest replacing “diagnostic performance” with “results”. Performance might be expected to include estimates of specificity, sensitivity etc. Could also include the >80% category in this table?

Paragraph from Line 274: could refer to the portable quantitative biosensor which has shown promising performance

Line 280: “increasingly insensitive”: the RDT is developed with an analytical threshold around 30%; individuals >30% activity are not intended to be identified by this assay

Discussion: occasional mention of the variant “class”: not sure what value specifying these classes adds, but may cause confusion/require some introduction for unfamiliar readers. Aren’t referred to elsewhere in the paper.

Figure 1: “quantitative” – check spelling in 3rd box down; suggest extending the flowchart to present the genotyping and subsequent seq steps to illustrate the ND category

Figure 3: really interesting figure! Suggest including the 80% cut-off line to emphasise that the plot represents only results below this levels. Could also add N values to the x-axis labels

Fig 4: Another great figure. Also suggest adding 80% activity line. Could consider including a dotted/hash overlay to the homo/compound variants so that they are more clearly differentiated from the hetero samples?

Line 400 (ref 18): “eeficiency”

**Summary and General Comments**

Reviewer #1: The paper provides a timely analysis of the relative advantages of a 70% cut-off vs. an 80% G6PD cutoff. This analysis is urgently needed as the higher threshold will exclude a substantial number of vivax malaria patients from the benefits of the radical, exposes them to the risk of recurrent infections and reduces the chance of speedy elimination of vivax malaria. Clearly stating the negligible benefits of the higher threshold – as the paper does - is therefore a great service to the malaria community and as such most welcomed.

Reviewer #2: Kindly see above

Reviewer #3: In this study, Satyagraha and colleagues address a key difficulty with radical cure of P. vivax, namely how to diagnose eligibility for safe treatment. G6PD phenotypes are a continuous spectrum, and where to impose the thresholds for treatment is a trade-off between potential haemolytic events and limiting access to treatment which could protect patients against relapse. The data collected in this study very clearly presents this. The study draws on three different classes of diagnostics to investigate this question in a high-endemic island of eastern Indonesia. I really enjoyed reading this paper, however I have questions about the potential impact of how the 100% normal activity level was defined, and whether this has implications for the application of the conclusions to different contexts.

Main comments:

Variable spectrophotometry across laboratories has meant that relative activities (as percentages) rather than absolute G6PD levels have been used to define thresholds for normal vs. deficient activity, e.g. with eligibility for tafenoquine set to 80% of normal levels.

A widely adopted convention for defining 100% activity is the “Adjusted Male Median”, which is based on male activity levels. Readers who do not review the Methods section of this paper will assume that this is what was followed here too. Given that this study screened only females, the 100% level was defined as the mean activity among those with >= 5U/gHb. Given the influence that this 100% value has on the study’s analysis and conclusions (e.g. advantages of 70% vs 80%), the authors should provide more rationale and justification (How was 5U/gHb selected? How does the 100% absolute value compare to previously defined AMMs from this lab/population?) Authors should discuss the implications of this definition, and whether their conclusions regarding the 70 vs 80% cut-off would apply more broadly. The limitations of this system of defining study-specific relative thresholds could be discussed. In the Discussion, consider specifying “threshold of 70% of normal, as defined in this study, …” (line 262).

Discussion of the Carestart RDT performance should be in the context of the test’s analytical threshold being around the 30% level. It is designed to not distinguish 30-80% from >80% individuals. (e.g. lines 106-107, 280).

Introduction should discuss the clinical risks of radical cure to heterozygotes with 30-80% activity levels.

Methods: how were study participants recruited? Were genetically related individuals excluded? (e.g. sisters or mother/daughters?)

Results: I don’t think the authors report on whether any other variants beyond those in the SNP panel were identified by the gene sequencing. E.g. could just specify if these results will be published elsewhere.

PLOS authors have the option to publish the peer review history of their article (what does this mean?). If published, this will include your full peer review and any attached files.

Reviewer #1: No

Reviewer #2: No

Reviewer #3: No
---

## [Editor Report · Decision Letter 1]

30 Jun 2021

Dear Dr Satyagraha,

We are pleased to inform you that your manuscript 'Genotypes and Phenotypes of G6PD Deficiency Among Indonesian Females Across Diagnostic Thresholds of G6PD Activity Guiding Safe Primaquine Therapy of Latent Malaria' has been provisionally accepted for publication in PLOS Neglected Tropical Diseases.

Best regards,

Wuelton Marcelo Monteiro, Ph.D.

Associate Editor

Mary Lopez-Perez

Deputy Editor

---

## [Editor Report · Acceptance letter]

13 Jul 2021

Dear Dr Satyagraha,

We are delighted to inform you that your manuscript, "Genotypes and Phenotypes of G6PD Deficiency Among Indonesian Females Across Diagnostic Thresholds of G6PD Activity Guiding Safe Primaquine Therapy of Latent Malaria," has been formally accepted for publication in PLOS Neglected Tropical Diseases.

Best regards,

Shaden Kamhawi

co-Editor-in-Chief

Paul Brindley

co-Editor-in-Chief
